# Effects of a Blend of Live Yeast and Organic Minerals as an Alternative to Monensin on Intake, Digestibility, Performance and Beef Quality of Nellore Bulls Finished on Pasture with High Concentrate Supplementation

Maxwelder Santos Soares [1], Luis Henrique Curcino Batista [1,*], Ivanna Moraes Oliveira [2], Hugo Aparecido Silveira Issa [2], Iorrano Andrade Cidrini [1], Igor Machado Ferreira [1], Luiz Fernando Costa e Silva [3], Anne Koontz [4], Vaughn Holder [4], Gustavo Rezende Siqueira [1,2,*] and Flávio Dutra de Resende [1,2]

1 Department of Animal Science, Faculty of Agricultural and Veterinary Sciences, São Paulo State University (UNESP), Jaboticabal 14884-900, São Paulo, Brazil
2 Agência Paulista de Tecnologia dos Agronegócios (APTA), Colina 14770-000, São Paulo, Brazil
3 Alltech, Maringa 87030-405, Parana, Brazil
4 Alltech Inc., Nicholasville, KY 40356, USA
* Correspondence: luishenrique_94cb@hotmail.com (L.H.C.B.); siqueiragr@gmail.com (G.R.S.)

**Abstract:** Effects of a blend of live yeast and organic minerals as an alternative to monensin and inorganic trace minerals for beef cattle finished on pasture with high concentrate supplementation, on growth performance, intake, digestibility, liver trace mineral and carcass characteristics were evaluated. Forty-eight Nellore bulls were blocked by initial body weight and randomly assigned to one of the two experimental diets. The animals were raised in an experimental pasture divided into 12 paddocks equipped with an electronic system for monitoring individual feeding behavior and feed intake. Treatments were: (1) Monensin (MON), 30 mg/kg supplement dry matter of sodium monensin and trace minerals supplementation from inorganic sources; (2) AdvantageTM (ADV), 1.6 g/kg supplement DM of a blend of live yeast (*Saccharomyces cerevisiae* strains) and organic trace minerals. The use of ADV instead of MON led to greater supplement intake and greater intake of dietary components. Bulls fed ADV also had higher digestibility of dry matter, organic matter, crude protein and non-fiber carbohydrates. Bulls fed MON had a greater number of visits to the feeder, however with a shorter time spent feeding per visit. The use of ADV resulted in higher average daily weight gain, and feed efficiency was similar between treatments. In the carcass, ADV tended toward greater Longissimus muscle area. Liver Zn concentration tended to be lower in the ADV treatment. The use of ADV generated higher meat lightness and redness. In summary, the blend of live yeast and organic minerals can be an alternative to monensin and inorganic sources of trace minerals for beef cattle finished on pasture with high concentrate supplementation, without negative effects on supplement feed efficiency and with benefits to animal growth.

**Keywords:** carcass; chromium supplementation; ionophore; nutrition; yeast-based blend

## 1. Introduction

Technologies in feed additives are of interest in ruminant nutrition, to promote health, growth, feed efficiency and improvement on carcass characteristics, thus bringing greater productivity to livestock activity. Sodium monensin is an ionophore that has been used to improve feed efficiency and decrease the risk of acidosis in finishing beef cattle [1,2]. However, there is a growing demand from consumers for meat produced free from the use of antibiotics, due to the public concern of increasing antimicrobial resistance in people.

Several non-antimicrobial products have been evaluated as feed additives, especially for cattle fed high-concentrate diets, in order to replace the use of monensin without impairing feed efficiency and rumen health [3,4]. Yeast-based products have been studied for

this purpose, however with variable results [3,5]. Batista et al. [6], in a meta-analysis study considering several yeast-based feed additives, reported that the main benefits of yeast products for beef cattle were in improving digestibility and rumen health, and generating greater average daily gain (ADG). However, the authors emphasize that the benefits in animal performance may be present in low magnitudes [6], requiring investments in new technologies in yeast products, such as research on specific strains and combination with other technologies, in order to achieve greater benefits in animal performance.

A topic that has been of increasing interest in cattle supplementation is the use of organic complexed sources of Co, Cu, Mn, Se and Zn in place of inorganic sources [7]. This is in addition to the supplementary use of Cr, which was established as essential by NASEM [8], however without established requirements. In this context, organic sources of trace minerals have been recognized for showing greater absorption, retention and biological activity compared to inorganic minerals [9–11], thus enabling the reduction of supplementary levels without harming the mineral status during a finishing period. Vellini et al. [12] observed greater feed efficiency and Longissimus muscle area (LM area) in feedlot-finished Nellore bulls with supplementation of zinc amino acid complex in association with chromium methionine, compared to mineral supplementation from inorganic sources including zinc, but without chromium. In addition, the use of a blend of live yeast and organic minerals instead of monensin or a diet without feed additives increased milk and solids yield of dairy cows during the hot season [13].

Because of the possible improvement on digestibility, rumen health and performance, with the use of live yeast and the positive impacts of organic sources of trace minerals, including chromium, on feed efficiency and carcass characteristics, we hypothesized that the combination of technologies, such as specific strains of live yeast + organic minerals, could replace the traditional use of monensin and inorganic sources of trace minerals, during the finishing phase of beef cattle. Therefore, the objective of this experiment was to evaluate the effects of a blend of live yeast and organic minerals as an alternative to monensin and inorganic trace minerals on intake, digestibility, performance and beef quality of Nellore bulls finished on pasture with high concentrate supplementation.

## 2. Materials and Methods

All experimental procedures in this study were carried out in accordance with the ethical principle established by the Brazilian Council for the Control of Animal Experimentation and approved by the Ethics Committee for the Use of Animals of the Sao Paulo State University, College of Agricultural and Veterinary Sciences, Sao Paulo, Brazil (process no. 014782/19).

### 2.1. Experimental Area, Animals, Treatments

This experiment was conducted from June to October 2018 at the Research Center of APTA (Agência Paulista de Tecnologia dos Agronegócios) in Colina, São Paulo, Brazil. The experimental period comprised the dry season in central Brazil (20°43′5′′ S, 48°32′38′′ W), which is characterized by low rainfall and low pasture quantity and nutritional quality. The area was an experimental pasture, consisting of Marandu grass (*Brachiaria brizantha* cv. Marandu) implanted at the beginning of the rainy season of 2015, and divided into 12 paddocks of 1 ha each. The paddocks were equipped with an electronic system for monitoring individual feeding behavior and feed intake (Intergado Ltd., Contagem, Minas Gerais, Brazil; [14]). Each paddock was grazed by four animals (Two/treatment), with individual access to the electronic feeder for each animal. The electronic feeders allow the intake of concentrate by only one animal per specific feeder, allowing the application of treatments with different supplementation strategies and/or different feed additives, with the animals being in the same pasture, therefore the same grazing conditions.

In total, 52 Nellore bulls (BW = 519 $\pm$ 25 kg, 33 $\pm$ 3 months) distributed in a randomized complete block design were used. Animals were stratified by initial body weight (initial BW) and distributed to the grazing area. Four animals, representative of different

BW blocks, were slaughtered at the beginning as the baseline group to represent the initial carcass weight of the remaining animals ($n = 48$). All animals originated from the same research center and had the same feed and health management throughout the growth phase, until the beginning of the experiment, being raised in a pasture area with free access to water and mineral supplement from inorganic sources.

The experimental period lasted 112 days, divided into four periods of 28 days. This period comprised the adaptation phase to concentrate intake and the finishing phase of the bulls on pasture, where high concentrate supplementation was provided (Table 1). This system of finishing cattle on pasture has been practiced in Brazil with satisfactory results in terms of productivity and meat quality [15,16]. Adaptation to concentrate intake occurred in the first 28 days (first experimental period). It started with the offer of concentrate at 1% of the initial BW, and every three days the offer was increased by 0.25% of the BW, until reaching the equivalent of 2% of the BW, after which concentrate was offered ad libitum.

**Table 1.** Ingredient and nutrient composition of supplements.

| Item (g/kg of DM) | MON [1] | ADV [2] |
|---|---|---|
| Cracked corn | 802 | 802 |
| Citrus pulp | 100 | 100 |
| Peanut meal | 60 | 60 |
| Urea | 10 | 10 |
| Optigen® | 5.0 | 5.0 |
| Macromineral mix [3] | 22.3 | 22.3 |
| Inorganic trace mix [4] | 0.126 | - |
| Advantage[TM] [5] | - | 1.60 |
| Rumensin 200 | 0.153 | - |
| Chemical Analysis, g/kg of DM | | |
| Crude protein | 127 | 127 |
| Neutral detergent fiber | 158 | 158 |
| Ash | 25.7 | 25.7 |
| Ether extract | 34.9 | 34.9 |
| Total carbohydrate | 775 | 775 |
| Non-fibrous carbohydrate | 616 | 616 |

[1] Supplement with monensin (30 mg/kg of DM) and trace minerals from inorganic sources. [2] Monensin-free supplement, with a blend of live yeast and organic minerals. [3] Containing (Supplement DM basis) 6.5 g/kg limestone, 5.0 g/kg dicalcium phosphate, 1.1 g/kg sulfur, 1.1 g/kg magnesium oxide, 4.3 g/kg potassium chloride, 4.3 g/kg salt. [4] Supplemental trace minerals per kg of supplement: 0.68 mg Co, 12 mg Cu, 1 mg I, 15.01 mg Mn, 45.6 mg Zn, 0.18 mg Se. Sources of trace mineral included copper sulfate, zinc oxide, manganese monoxide, calcium iodate, sodium selenite, and cobalt sulfate. [5] Feed additive with live yeast (*Saccharomyces cerevisiae* strains 1026 + 8417; amount of $3.2 \times 10^8$ cfu/g in the final supplement) and organic sources of trace minerals. Supplemental trace minerals per kg of supplement: 0.14 mg Co, 5.34 mg Cu, 0.67 mg I, 10.62 mg Mn, 6.40 mg Fe, 16.0 mg Zn, 0.12 mg Se, 0.53 mg Cr. Sources of trace mineral included copper proteinate, manganese proteinate, zinc proteinate, iron proteinate, selenium-enriched yeast, and chromium yeast.

The experimental treatments were (1) Monensin (MON), 30 mg/kg supplement DM of sodium monensin (Rumensin; Elanco) and trace minerals supplementation from inorganic sources at the following rates (mg/kg) in concentrate: 0.68 mg Co, 12 mg Cu, 1 mg I, 15.01 mg Mn, 45.6 mg Zn, 0.18 mg Se. Inorganic trace mineral sources included copper sulfate, zinc oxide, manganese monoxide, calcium iodate, sodium selenite, and cobalt sulfate; it was formulated to meet or exceed NASEM [8] recommendations. (2) Advantage[TM] (ADV), 1.6 g/kg supplement DM of Advantage (Alltech®). Advantage[TM] is a blend of live yeast (*Saccharomyces cerevisiae* strains 1026 + 8417; $3.2 \times 10^8$ cfu/g) and organic minerals supplemented at the following rates (mg/kg) of concentrate: 0.14 mg Co, 5.34 mg Cu, 0.67 mg I, 10.62 mg Mn, 6.40 mg Fe, 16.0 mg Zn, 0.12 mg Se, 0.53 mg Cr. Organic trace mineral sources included copper proteinate, manganese proteinate, zinc proteinate, iron proteinate, selenium-enriched yeast, and chromium yeast.

Monensin inclusion level was in accordance with Lemos et al. [17]. Advantage[TM] was provided following manufacturer's recommendations (Expected daily intake of 3 g/100 kg

of BW). Both additives were provided mixed with the concentrate. Concentrate supplement were formulated to meet or exceed NASEM requirements [8] for protein and energy, to an average daily gain of 1.5 kg (Table 1). The concentrate supplement was provided once daily (08:00 h) in quantity for ad libitum intake, with an expected intake of approximately 2% of BW. The amount of concentrate offered was adjusted when the leftovers were greater than 10% for four consecutive days; this criterion was established to avoid variations in the daily dry matter intake.

## 2.2. Pasture Characteristics

Every paddock received all the treatments; therefore, the grazing conditions were similar across treatments. Quantitative and qualitative assessments of the pasture conditions, measuring forage mass, structural components, and chemical composition, were performed (Table 2) at the beginning of the experiment and every 28 days. The forage mass was estimated using the double sampling method [18]. The quantitative and structural components of the forage sward were evaluated at medium heights. Forage samples were separated into green leaf, dead leaf, green stem, and dead stem. After separation, forage components were weighed and oven-dried at 55 °C for 72 h to obtain the partial DM and the proportion of each component in the forage sward (Table 2). Hand-plucked samples were used to estimate the dietary nutritional value [19]. Samples were oven-dried at 55 °C for 72 h and then ground in a Wiley mill (Thomas Model 4, Thomas Scientific, Swedesboro, NJ, USA) to pass a 1 and 2-mm mesh sieve, for further analysis. Samples were analyzed for neutral detergent fiber (NDF), acid detergent fiber (ADF), organic matter (OM), lignin, ether extract (EE), non-fibrous carbohydrate (NFC) and nitrogen (N).

**Table 2.** Characteristics of the forage sward, grazed by Nellore bulls finished on pasture with high concentrate supplementation, with monensin (MON) or a blend of live yeast and organic minerals (ADV).

| Item | Periods | | | | | | SEM |
|---|---|---|---|---|---|---|---|
| | d 0 | d 28 | d 56 | d 84 | d 112 | | |
| Quantitative and structural characteristics | | | | | | | |
| Sward height, cm | 38.4 | 30.7 | 28.5 | 26.5 | 29.0 | | 2.30 |
| Forage mass, kg DM/ha | 5946 | 4331 | 3961 | 3271 | 3802 | | 508 |
| Senescent leaf, g/kg DM | 208 | 210 | 190 | 270 | 80.0 | | 35.0 |
| Senescent stem, g/kg DM | 694 | 790 | 810 | 730 | 750 | | 23.0 |
| Green leaf, g/kg DM | 4.03 | 0.0 | 0.0 | 0.0 | 120 | | 27.0 |
| Green stem, g/kg DM | 94.0 | 0.0 | 0.0 | 0.0 | 50.0 | | 21.0 |
| Chemical composition, g/kg DM | | | | | | | |
| Organic matter | 943 | 934 | 956 | 955 | 947 | | 4.80 |
| Neutral detergent fiber | 830 | 839 | 840 | 841 | 717 | | 2.51 |
| Acid detergent fiber | 499 | 505 | 502 | 505 | 384 | | 1.40 |
| Lignin | 90.8 | 92.9 | 91.1 | 97.0 | 52.7 | | 1.41 |
| Crude protein | 34.2 | 32.8 | 32.9 | 27.9 | 79.6 | | 1.46 |
| Ether extract | 9.92 | 7.55 | 6.02 | 6.41 | 11.83 | | 0.82 |
| Non-fibrous carbohydrate | 68.4 | 54.7 | 77.0 | 77.5 | 139 | | 5.25 |

In vitro dry matter digestibility (IVDMD).

## 2.3. Feed Intake and Digestibility

Individual concentrate supplement intake was monitored daily throughout the experimental period, by calculating the difference between the DM of concentrate offered and refusals. The behavior of concentrate supplement intake in terms of time spent feeding per day, number of visits per day to the feeder, and time spent feeding per visit were monitored daily by the electronic system for monitoring individual feeding behavior and feed intake. The mean per animal per period was considered later for statistical analysis.

Forage intake was determined in 24 animals randomly selected from the 48 experimental animals. To determine fecal excretion, the external marker, titanium dioxide ($TiO_2$) and the internal marker, indigestible NDF (iNDF) were used [20]. Estimates of forage intake and total tract digestibility were evaluated in the second and fourth experimental periods (d 28 to 56 and d 84 to 112). The $TiO_2$ was offered during the last 10 days of the respective experimental period at 10 g/animal/d. Six days of supply aimed at establishing the marker's fecal flow, and the last four days to collect fecal samples, respectively, at: 1700, 1300, 1000, and 0700 h, to represent fecal excretion during the day [21]. Feces were pooled by animals within period, dried to a constant weight at 55 °C, ground as described above, and stored for analysis of chemical composition and $TiO_2$. On the same days of fecal collections, a hand-plucked forage sample was used to quantify the iNDF of forage.

Forage DMI was estimated as previously described by Miorin et al. [22], where the fecal output of the internal marker (iNDF) was corrected for the contribution of the concentrated supplement as follows:

$$\text{Forage DMI} = \text{FE} \times [\text{iMF}] - \text{DMIS} \times [\text{iMS}]/[\text{iMH}]$$

where FE is the fecal excretion, DMIS is the DMI of supplement, [iMF], [iMS] and [iMH] are the concentrations of the internal marker in feces, supplement, and forage, respectively. Total DMI was obtained by addition of forage and supplement DMI. The total tract apparent digestibility was estimated using the following model:

$$\text{DM digestibility} = (\text{DMI} - \text{FE})/\text{DMI} \times 100$$

Digestibility of DM, OM, CP, NDF, EE and NFC were calculated based on the intake and fecal excretion of individual components.

*2.4. Animal Performance and Carcass Assessments*

Bulls were weighed at the start (0 d) of the experiment, and at the end of each experimental period, after fasting for 16 h (feed and water). Average daily gain (ADG) was calculated for each experimental period. Concentrate supplement feed efficiency (supplement G:F) was calculated by dividing the ADG by the supplement intake.

As mentioned above, four animals were slaughtered at the beginning of the experiment (randomly selected within the BW blocks, to estimate the initial carcass weight of the remaining animals. Animals were transported to the slaughterhouse (Minerva Foods, Barretos, SP, Brazil) located 20 km from the research facility. After arrival at the slaughterhouse, animals were kept in resting pens for 18 h (free access to water) and then submitted to humanitarian slaughter under Brazilian Federal Inspection, and the hot carcass weight (HCW) was obtained. The hot carcass weight (HCW) was measured after evisceration without kidneys, pelvic, and heart fat. Carcass gain was calculated using a linear equation to predict the initial HCW. The equation was applied to the initial BW (kg) to determine a predicted initial HCW (kg), as follows:

$$\text{HCW}_{\text{initial}} = -22.866 + 0.6038 \times \text{BW}_{\text{initial}} \ (R^2 = 0.99)$$

At the end of the evaluation period, all remaining animals were slaughtered following the same procedure as that of the baseline groups, and the final HCW was obtained. Subsequently, the carcasses were placed in a cold chamber at 2 °C for 24 h. Carcass-adjusted gain was determined by subtracting the final HCW from the initial HCW dividing the results by the days on evaluation period. Twenty-four hours after slaughter, ribeye area and backfat thickness were measured on the left side of the carcass, between the 12th and 13th ribs [23]. Twenty-four hours after slaughter and cooling, four 2.54 cm-thick steaks from *longissimus thoracis* muscle were collected from the left side of the carcass from the 13th rib toward the head for chemical composition, color, Shear Force and cooking loss analyses. Each steak was identified, and vacuum packed in polyethylene bags (water vapor

permeability < 10 g/m$^2$/24 h at 38 °C and oxygen permeability < 40 mL/m$^2$/24 h at 25 °C) and stored at −20 °C until further analysis.

### 2.5. Liver Samples

Liver samples were collected on the day of slaughter from the left lobe of each liver after being inspected by the Brazilian Federal Inspection personnel. Each sample was placed in numbered cryotubes corresponding to the carcass order, placed on ice and transported to the Research Center Laboratory and stored at −80 °C until analyzed for trace mineral concentrations. Liver samples were collected only at slaughter: (1) because our goal was to evaluate the final liver trace mineral concentrations of bulls after receiving the supplementation during the finishing period, and (2) to avoid a surgery-induced inflammatory response in the beginning of the study that could interferes with growth performance and physiological parameters [24]. It is worth mentioning that before the beginning of the experiment the animals were raised in the same group, grazing and mineral supplementation conditions, therefore the possibility of variation between animals in the initial liver trace mineral concentrations, were randomly distributed among the treatments at the beginning of the experiment.

### 2.6. Laboratory Analyzes

Samples of concentrates, forage and feces previously ground to 1 mm were analyzed for contents of DM (method 930.15; [25]), CP (N × 6.25; Kjeldahl method 984.13; [25]), ether extract (method 920.39; [25]), ADF and lignin (method 973.18; [25]), ash (method 942.05; [25]), and NDF using α-amylase [26]. The NFC content was calculated according to Hall et al. [27] as NFC = 1000 − [(NDF − NDIP) + CP + ether extract + ash], with values expressed as grams per kilogram of DM. The indigestible NDF of forage, concentrates, and feces was determined by an in-situ incubation procedure for 288 h [20]. After incubation, bags were removed from the rumen, washed in running tap water until bleaching, and subjected to NDF analysis as previously described. The determination of TiO$_2$ concentrations in feces was performed through spectrophotometry read at 410 nm as described by Myers et al. [28]. Liver samples were sent to a commercial laboratory (Bio Minerais Análises Químicas Ltda, Campinas, São Paulo, Brazil) for analysis of the concentrations of Cu, Mn, Se and Zn using inductively coupled plasma atomic emission spectrometry–ICAP 6300–Thermo scientific.

For the proximate analysis, steaks were thawed overnight at room temperature (4 °C), ground and used to analyze composition (protein, ether extract, ash and moisture) using near infrared analyses (AOAC method: 2007–04) using a Foodscan™ (FOSS, Hillerod, Denmark). The cooking loss was performed according to Aroeira et al. [29]. The steaks were weighed and grilled at 160–180 °C until their center reached an internal temperature of 71 °C [30]. After temperature stabilization, the steaks were weighed, and cooking loss was calculated as a percentage of the weight of the steaks before the cooking process. Shear force was analyzed using the Warner–Bratzler square shear force method. Six rectangular core samples (1.0 × 1.0 × 2.5 cm) were manually removed from each steak parallel to the muscle fibers. The core was completely sheared perpendicular to the muscle fibers in the Texture Analyzer (Brookfield, model CT325K, Middleboro, MA, EUA) with Warner–Bratzler blade of a 1.016 mm and 200 mm min–1 of speed. The maximum force (kg) was measured, and the average value was calculated for each steak.

The instrumental color analysis was performed on the surface of the steaks using a CM–700 spectrophotometric colorimeter and the CIELAB system with an 8 mm aperture, illuminant A, and 10° observer angle. Before the color readings, each steak was removed from the vacuum package and exposed to atmospheric air for 30 min for blooming and oxygenation of myoglobin. The lightness (L*), redness (a*), and yellowness (b*) components were recorded using an average of five readings per steak. The polar coordinates chroma (C*) and hue angle (h*) were also determined as: C* = [(a*)2 + (b*)2] × 0.5 and h* = tan − 1 (b*/a*).

### 2.7. Statistical Analysis

All data were analyzed as randomized complete block design using the MIXED procedures of SAS® University Edition software. Two bulls per treatment per paddock were the experimental unit, where bull(paddock) was included as random effect in all analyses. The variables when evaluated over the experimental periods (Supplement Intake, Supplement intake behavior, forage intake, digestibility, ADG and supplement G:F) were analyzed as repeated measures, and tested for fixed effects of treatment, time, and resulting interaction, using paddock(treatment) as the subject. Different covariance structures were tested with the final choice depending on lowest value for the Akaike information criterion. The variables that were not evaluated by period: Carcass and meat parameters, and liver trace minerals, were used in the model as a fixed effect only the effect of treatments. All results are reported as least squares means. Data were separated using PDIFF if a significant F-test was detected. Significance was set at $p \leq 0.05$, and tendencies were determined if $p > 0.05$ and $\leq 0.10$.

### 3. Results

Concentrate supplement intake, when measured during the periods of simultaneous evaluation of forage intake and apparent digestibility (d 28 to 56 and d 84 to 112), was 18.9% higher in animals receiving ADV ($p < 0.01$) compared to animals supplemented with monensin and inorganic sources of trace minerals (MON; Table 3). However, the forage intake was not different between treatments ($p = 0.73$), also there was no Treat (treatment) × Per (period) interaction ($p = 0.75$). There was an effect of the experimental period ($p < 0.01$) on the estimate of forage intake, where a lower intake was observed between d 28 to 56 (1.88 kg of DM/d) and higher intake observed in the final period (d 84 to 112; 2.37 kg of DM/d). The total DM intake was 14.6% higher in animals receiving ADV ($p < 0.01$), consequently there was a higher intake of OM, CP, EE, and NFC ($p \leq 0.01$), and a trend towards higher intake of NDF ($p = 0.10$). The use of ADV instead of MON also resulted in greater digestibility of DM, OM, CP ($p \leq 0.04$), and tended toward greater digestibility of NFC ($p = 0.06$), however there was no effect of treatments or Treat × Per interaction on digestibility of NDF ($p \geq 0.53$).

**Table 3.** Intake and digestibility of Nellore bulls finished on pasture with high concentrate supplementation, with monensin (MON) or a blend of live yeast and organic minerals (ADV).

| Item | Treatment [1] | | SEM | p-Value | | |
|------|------|------|------|------|------|------|
| | MON | ADV | | Treat | Per | Treat × Per |
| Intake, kg/d | | | | | | |
| Supplement | 8.50 | 10.49 | 0.41 | <0.01 | 0.27 | 0.90 |
| Forage | 2.15 | 2.10 | 0.10 | 0.73 | <0.01 | 0.75 |
| Total DM | 10.81 | 12.66 | 0.41 | <0.01 | 0.48 | 0.79 |
| OM | 10.45 | 12.26 | 0.40 | <0.01 | 0.52 | 0.78 |
| CP | 1.16 | 1.41 | 0.05 | <0.01 | 0.50 | 0.87 |
| NDF | 3.16 | 3.45 | 0.12 | 0.10 | 0.18 | 0.98 |
| EE | 0.370 | 0.415 | 0.01 | 0.01 | <0.01 | 0.40 |
| NFC | 5.84 | 7.17 | 0.28 | <0.01 | 0.28 | 0.88 |
| Digestibility, g/kg DM | | | | | | |
| DM | 625.6 | 672.9 | 16.3 | 0.03 | 0.03 | 0.80 |
| OM | 660.0 | 701.1 | 15.3 | 0.04 | 0.03 | 0.77 |
| CP | 525.0 | 607.4 | 25.6 | 0.02 | 0.01 | 0.45 |
| NDF | 527.6 | 528.2 | 14.8 | 0.97 | < 0.01 | 0.53 |
| EE | 764.7 | 769.0 | 20.9 | 0.87 | < 0.01 | 0.11 |
| NFC | 776.1 | 813.7 | 14.5 | 0.06 | 0.48 | 0.76 |

[1] Treatment, MON: Supplement with monensin (30 mg/kg of DM) and supplemental trace minerals from inorganic sources. Supplemental trace minerals per kg of supplement: 0.68 mg Co, 12 mg Cu, 1 mg I, 15.01 mg Mn, 45.6 mg Zn, 0.18 mg Se. ADV: Monensin-free supplement, with a blend of live yeast and organic minerals (Advantage; Alltech), live yeast (*S. cerevisiae* strains 1026 + 8417; 3.2 × 108 cfu/g). Supplemental trace minerals per kg of supplement: 0.14 mg Co, 5.34 mg Cu, 0.67 mg I, 10.62 mg Mn, 6.40 mg Fe, 16.0 mg Zn, 0.12 mg Se, 0.53 mg Cr.

Concentrate supplement intake, when evaluated throughout the experimental period (d 0–112) was 9.5% higher in animals receiving ADV instead of MON (Table 4). However, the intake was similar ($p = 0.27$) during the first experimental period (d 0–28), which involved the adaptation of the animals to a high concentrate diet. In this context, post-adaptation intake (d 29–112) was 12.2% higher in animals receiving ADV. The behavior of supplement intake in terms of time spent feeding per day, number of visits per day to the feeder, and time spent feeding per visit are shown in Figure 1. There was effect of the experimental period for all responses ($p < 0.01$). However, there was no Treat × Per interaction ($p \geq 0.19$). Regarding the treatment effects, it was observed that the animals supplemented with ADV had a lower number of visits to the feeder and a longer time spent feeding per visit ($p < 0.01$), with no significant change in the total time spent feeding per day ($p = 0.38$).

**Table 4.** Supplement intake and animal performance of Nellore bulls finished on pasture with high concentrate supplementation, with monensin (MON) or a blend of live yeast and organic minerals (ADV).

| Item | Treatment [1] | | SEM | *p*-Value |
|---|---|---|---|---|
| | MON | ADV | | |
| Supplement Intake, kg/d | | | | |
| Adaptation (d 0–28) | 7.51 | 7.77 | 0.19 | 0.27 |
| Post-adaptation (d 29–112) [2] | 9.20 | 10.48 | 0.39 | <0.01 |
| Total (d 0–112) [2] | 8.80 | 9.73 | 0.27 | <0.01 |
| Performance, live measures | | | | |
| Initial BW, kg | 518.6 | 518.4 | 7.1 | 0.97 |
| Final BW, kg | 635.0 | 647.7 | 8.9 | 0.22 |
| ADG adaptation, kg | 0.41 | 0.46 | 0.11 | 0.73 |
| ADG post-adaptation, kg [2] | 1.30 | 1.42 | 0.05 | 0.02 |
| ADG (d 0–112), kg [2] | 1.07 | 1.18 | 0.05 | 0.06 |
| G:F adaptation | 0.053 | 0.056 | 0.015 | 0.88 |
| G:F post-adaptation [2] | 0.148 | 0.142 | 0.006 | 0.47 |
| G:F d 0–112 [2] | 0.126 | 0.121 | 0.006 | 0.61 |

[1] Treatment, MON: Supplement with monensin (30 mg/kg of DM) and supplemental trace minerals from inorganic sources. Supplemental trace minerals per kg of supplement: 0.68 mg Co, 12 mg Cu, 1 mg I, 15.01 mg Mn, 45.6 mg Zn, 0.18 mg Se. ADV: Monensin-free supplement, with a blend of live yeast and organic minerals (Advantage; Alltech), live yeast (*S. cerevisiae* strains 1026 + 8417; $3.2 \times 10^8$ cfu/g). Supplemental trace minerals per kg of supplement: 0.14 mg Co, 5.34 mg Cu, 0.67 mg I, 10.62 mg Mn, 6.40 mg Fe, 16.0 mg Zn, 0.12 mg Se, 0.53 mg Cr. [2] Least-squared means are based on repeated measures.

Supplementation with ADV tended ($p = 0.06$) toward greater ADG (9.3% or 0.110 kg/d) considering the entire experimental period (Table 4). Although no Treat × Per interaction was observed ($p = 0.61$), ADG measured only in the first period (d 0–28) was a similar between the treatments ($p = 0.73$), while the post-adaptation ADG was higher ($p = 0.02$) in animals fed ADV (8.4% or 0.120 kg/d). However, the final BW was similar between treatments ($p = 0.22$). Additionally, there were no differences between treatment on measures of feed efficiency of concentrate intake (Supplement G:F; $p \geq 0.47$).

In the evaluation of animal performance based on carcass gain (Table 5), there were no differences in carcass ADG or final HCW ($p \geq 0.16$). Nevertheless, ADV fed bulls tended toward greater LM area (5.8%; $p = 0.07$), without a difference in subcutaneous fat thickness ($p = 0.92$).

Liver Cu, Mn, Se and Zn data are shown in Table 6. There were no treatment differences in liver concentrations of Cu, Mn and Se ($p \geq 0.27$). However, ADV supplementation tended ($p = 0.06$) to result in a lower concentration of Zn.

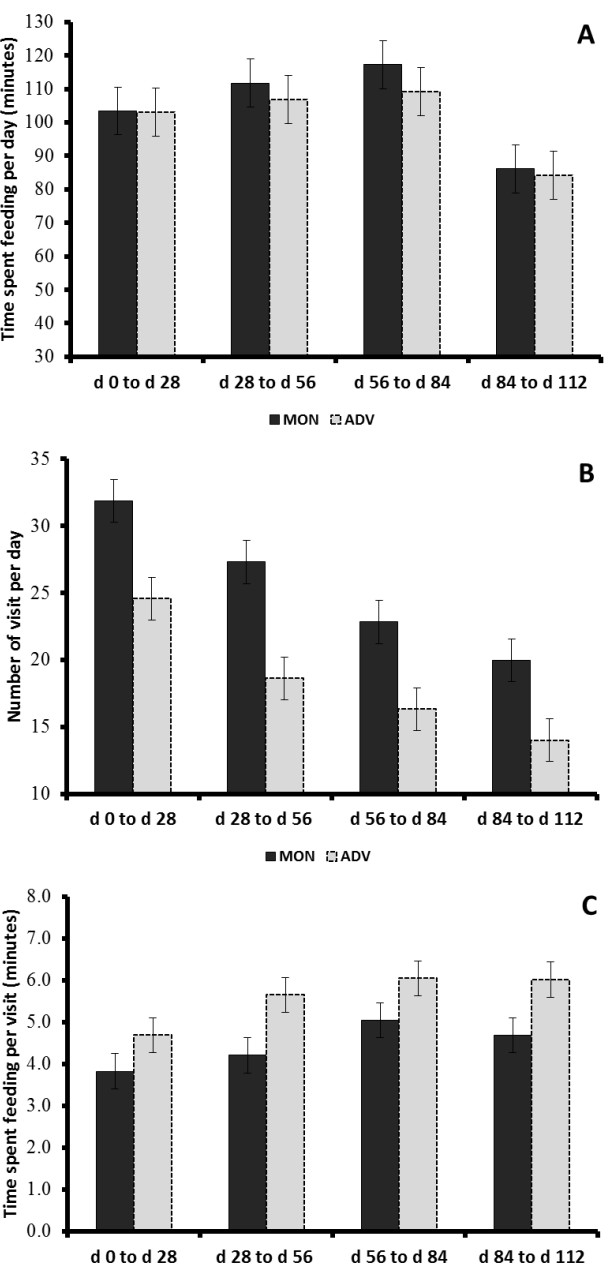

**Figure 1.** Supplement intake behavior of Nellore bulls finished on pasture with high concentrate supplementation, with monensin (MON) or a blend of live yeast and organic minerals (ADV). Time spent feeding per day (**A**), statistical effects: Period ($p < 0.01$), Treatment ($p = 0.38$), interaction Treat × Per ($p = 0.94$); Number of visits per day (**B**), statistical effects: Period ($p < 0.01$), Treat ($p < 0.01$), interaction Treat × Per ($p = 0.74$); Time spent feeding per visit (**C**), statistical effects: Period ($p < 0.01$), Treat ($p < 0.01$), interaction Treat × Per ($p = 0.19$).

In the evaluations of meat quality characteristics, there were no differences in meat proximate analysis ($p \geq 0.19$; Table 7), cooking loss ($p = 0.28$) or Warner–Bratzler Shear Force ($p = 0.69$). However, there was a tendency for ADV fed animals to have a greater carcass pH ($p = 0.08$). On meat colorimetric parameters, bulls fed ADV had higher L* and b* than those fed MON ($p \leq 0.03$).

**Table 5.** Carcass measures of Nellore bulls finished on pasture with high concentrate supplementation, with monensin (MON) or a blend of live yeast and organic minerals (ADV).

| Item | Treatment [1] | | SEM | *p*-Value |
|---|---|---|---|---|
| | **MON** | **ADV** | | |
| Initial HCW, kg | 290.3 | 290.1 | 4.2 | 0.95 |
| Final HCW, kg | 376.1 | 385.5 | 5.6 | 0.20 |
| Carcass ADG, kg | 0.77 | 0.84 | 0.03 | 0.16 |
| Supplement G:F | 0.090 | 0.088 | 0.004 | 0.70 |
| Dressing, % | 59.2 | 59.2 | 0.30 | 0.98 |
| Fat thickness, mm | 4.08 | 4.12 | 0.39 | 0.92 |
| LM area, cm$^2$ | 74.9 | 79.5 | 1.79 | 0.07 |

[1] Treatment, MON: Supplement with monensin (30 mg/kg of DM) and supplemental trace minerals from inorganic sources. Supplemental trace minerals per kg of supplement: 0.68 mg Co, 12 mg Cu, 1 mg I, 15.01 mg Mn, 45.6 mg Zn, 0.18 mg Se. ADV: Monensin-free supplement, with a blend of live yeast and organic minerals (Advantage; Alltech), live yeast (*S. cerevisiae* strains 1026 + 8417; $3.2 \times 108$ cfu/g). Supplemental trace minerals per kg of supplement: 0.14 mg Co, 5.34 mg Cu, 0.67 mg I, 10.62 mg Mn, 6.40 mg Fe, 16.0 mg Zn, 0.12 mg Se, 0.53 mg Cr.

**Table 6.** Liver concentrations of Cu, Mn, Se and Zn in Nellore bulls finished on pasture with high concentrate supplementation, with monensin (MON) or a blend of live yeast and organic minerals (ADV).

| Item | Treatment [1] | | SEM | *p*-Value |
|---|---|---|---|---|
| | **MON** | **ADV** | | |
| | Minerals | | | |
| Cu, mg/kg | 493.0 | 532.7 | 50.8 | 0.78 |
| Mn, mg/kg | 9.70 | 9.69 | 0.349 | 0.98 |
| Se, mg/kg | 1.32 | 1.23 | 0.064 | 0.27 |
| Zn, mg/kg | 174.2 | 157.6 | 5.18 | 0.06 |

[1] Treatment, MON: Supplement with monensin (30 mg/kg of DM) and supplemental trace minerals from inorganic sources. Supplemental trace minerals per kg of supplement: 0.68 mg Co, 12 mg Cu, 1 mg I, 15.01 mg Mn, 45.6 mg Zn, 0.18 mg Se. ADV: Monensin-free supplement, with a blend of live yeast and organic minerals (Advantage; Alltech), live yeast (*S. cerevisiae* strains 1026 + 8417; $3.2 \times 108$ cfu/g). Supplemental trace minerals per kg of supplement: 0.14 mg Co, 5.34 mg Cu, 0.67 mg I, 10.62 mg Mn, 6.40 mg Fe, 16.0 mg Zn, 0.12 mg Se, 0.53 mg Cr.

**Table 7.** Meat characteristics of Nellore bulls finished on pasture with high concentrate supplementation, with monensin (MON) or a blend of live yeast and organic minerals (ADV).

| Item | Treatment [1] | | SEM | *p*-Value |
|---|---|---|---|---|
| | **MON** | **ADV** | | |
| pH | 5.58 | 5.69 | 0.04 | 0.08 |
| | Proximate analysis | | | |
| Moisture, g/kg | 744 | 741 | 2.10 | 0.34 |
| CP, g/kg | 229 | 229 | 1.02 | 0.73 |
| EE, g/kg | 16.8 | 19.4 | 1.60 | 0.19 |
| Ash, g/kg | 10.3 | 10.4 | 0.11 | 0.54 |
| | Colorimetric parameters | | | |
| Ligthness (L*) | 35.7 | 36.9 | 0.43 | 0.03 |
| Redness (a*) | 17.0 | 17.5 | 0.33 | 0.25 |
| Yellowness (b*) | 12.5 | 13.8 | 0.25 | <0.01 |
| Chroma (C*) | 21.2 | 22.0 | 0.44 | 0.21 |
| Hue angle (h*) | 36.6 | 37.2 | 0.29 | 0.11 |
| Cooking loss, % | 31.2 | 32.5 | 0.85 | 0.28 |
| WBSF, (N) [2] | 54.0 | 52.4 | 2.94 | 0.69 |

[1] Treatment, MON: Supplement with monensin (30 mg/kg of DM) and supplemental trace minerals from inorganic sources. Supplemental trace minerals per kg of supplement: 0.68 mg Co, 12 mg Cu, 1 mg I, 15.01 mg Mn, 45.6 mg Zn, 0.18 mg Se. ADV: Monensin-free supplement, with a blend of live yeast and organic minerals (Advantage; Alltech), live yeast (*S. cerevisiae* strains 1026 + 8417; $3.2 \times 108$ cfu/g). Supplemental trace minerals per kg of supplement: 0.14 mg Co, 5.34 mg Cu, 0.67 mg I, 10.62 mg Mn, 6.40 mg Fe, 16.0 mg Zn, 0.12 mg Se, 0.53 mg Cr.
[2] WBSF = Warner–Bratzler Shear Force.

## 4. Discussion

Sodium monensin is the feed additive that has been widely used for beef cattle finished in feedlots in Brazil [2]. Using a meta-analytic approach, Duffield et al. [1] demonstrated that the use of monensin in growing and finishing beef cattle diets can improve animal performance by reducing DMI (3%) and increasing both ADG (2.5%) and feed efficiency (6.4%). According to Tedeschi et al. [31], the main effects of feeding monensin to ruminants include reduction of feed intake, inhibition of ruminal protein degradation, thus increasing protein escape from the rumen, and an increase in propionate and decrease in methane production in the rumen. Our hypothesis was that a new feed additive, combining live yeast and organic trace minerals, would support performance in finishing beef cattle receiving high concentrate supplementation, without the use of monensin. In fact, a recent meta-analysis demonstrated the benefits of using yeast-based products for beef cattle fed high-grain diets, in terms of nutrient digestibility, rumen health, and gains in DMI, ADG and FE, however, the performance benefits may be of low magnitude [6]. In this context, the authors suggest that new technologies in yeast products, in terms of specific strains and combinations with other technologies, can be developed in order to complement the current benefits and obtain greater gains in animal performance [6].

The higher concentrate intake (9.5%) in animals receiving ADV, in relation to the MON treatment, may be related to the expected reduction in intake with the use of monensin in relation to a diet without feed additives [1]. This is in addition to a possible stimulus to consumption with the use of ADV, due to the action of yeast benefitting rumen health [6]. Monensin has been responsible for changes in the rumen microbiota, greater propionate production and changes in satiety mechanisms, which can result in a reduction in the amount of feed ingested and an increase in the frequency of meals [32]. In fact, in our data regarding the behavior of supplement intake, it was observed that animals receiving MON had a higher number of visits per day to the supplement and a shorter time spent feeding per visit. However, the total time spent feeding per day was not different between MON and ADV, due to the fact that although ADV generated fewer visits, the time spent feeding also increased, a behavior that, associated with higher supplement intake in this treatment, indicates that the animals in ADV had a higher consumption of concentrate per visit to the feeder. This behavior could favor the occurrence of metabolic disorders associated with high concentrate intake, such as the occurrence of subacute ruminal acidosis (SARA) [33]. However, the high intake of concentrate in ADV and the good animal performance in this treatment indicates that the feed additive was able to maintain the rumen health of the bulls, since one of the main problems in the occurrence of SARA is irregular intake and reduced performance [33]. Yet, studies evaluating rumen pH and fermentative profile using the present blend of yeasts and organic minerals are necessary to test this hypothesis.

During the periods of evaluation of forage intake and digestibility (d 28 to 56 and d 84 to 112), greater intake of concentrate supplement in ADV was also observed. However, the forage intake was not different between treatments, demonstrating that within the production model of this study, the results comparing feed additives reflect only on the consumption of the concentrate. Few studies to date have evaluated forage intake by cattle finished on pasture with high concentrate supplementation. Simioni et al. [15] evaluated pasture intake in beef cattle of different genotypes (Nellore, $\frac{1}{2}$Angus and $\frac{1}{2}$Senepol) finished with supplementation level similar to that adopted in the present study, and observed that the participation of forage in the diet ranged from 6.26 to 14.5%. In the present study, the proportion of forage in the diet ranged from 16.5 to 19.8% in ADV and MON, respectively. The difference observed between the experimental periods on forage intake may be related to the occurrence of rain during the final experimental period, which led to the emergence of green leaves available for grazing, which may have favored the ingestion of pasture in the final evaluation period.

The evidence in this study indicates that the greater total DM intake observed with the use of ADV in relation to the use of MON, can be attributed to the greater intake of concentrate. Consequently, the consumption of diet components (OM, CP, NDF, NFC) was

higher in ADV. Shen et al. [34] also reported higher intake of DM, OM, NDF and starch in finishing beef heifers with the use of a *Saccharomyces cerevisiae* fermentation product compared to the use of antibiotics (monensin + tylosin). In addition to the higher intake, the use of ADV generated greater digestibility of DM, OM, CP and tended toward greater NFC digestibility. In fact, one of the most consistent benefits of using yeast products has been the improvement in diet digestibility [6,35,36]. Many of these benefits have been attributed to an improvement in the rumen environment, where the addition of yeast can stimulate specific ruminal microorganisms [37]. Increased concentrations of ruminal fibrolytic bacteria have been observed using yeast supplementation [38,39]. In this context, although it is frequent in studies using yeast products to observe an effect on NDF digestion [6,35,36], in the present study, the use of ADV did not affect NDF digestibility. In fact, Batista et al. [6] observed, in the meta-analysis study, high heterogeneity between studies in NDF digestibility responses and attributed, as a possible source of heterogeneity, factors such as the type of forage used, the amount of fiber in the diet, as well as the use of fiber-rich co-products. In the present study, the lack of effect on NDF digestibility is probably related to the low concentration of NDF in the concentrate, consequently little NDF ingested from it and the low quality of the fiber from the pasture.

The higher average daily gain observed with the use of ADV in the post-adaptation periods and the tendency for greater ADG when evaluated throughout the experimental period may be associated with the greater concentrate supplement intake in these periods, as well as the greater digestibility of nutrients observed with the use of ADV. In the same way that ADG was similar between treatments in the first experimental period, concentrate supplement intake was not different between treatments. It is worth mentioning that in this period there was a gradual increase in the supply of concentrate for 15 days in order to adapt the animals to the high-concentrate diet. Therefore, there was no ad libitum offer of the concentrate during this period. Thus, with the limited supply of concentrate in part of the period, associated with changes in rumen dynamics, passage rate, and consequently a change in the amount of gastrointestinal content, due to the transition from a diet high in forage to a diet with high concentrate proportion [40], may have influenced the possibility of observing the effect of treatments on ADG measurement in this first period.

According to our knowledge, so far, this was the first study to evaluate the use of an additive based on live yeasts and organic minerals as an alternative to monensin for beef cattle finished on pasture with high concentrate supplementation. Batista et al. [6] reported greater ADG in beef cattle with the use of several yeast products, however, in magnitudes lower than those observed in the present study (0.036 kg/d or 2.3% in relation to the study database). Likewise, the use of technologies in organic minerals has presented variable responses in performance and health of beef cattle [7]. Nevertheless, Dorton et al. [41] reported increased feedlot receiving ADG when beef cattle were supplemented with organic complexed Zn, Cu, Mn and Co during 30 days post-weaning and 28 days post receiving of feeder cattle. In this context, the combination of technologies, such as live yeasts + organic sources of trace minerals used in ADV, showed benefits of higher magnitudes on ADG of Nellore cattle under the conditions of the present study. The use of a blend of live yeast and organic minerals instead of monensin or a diet without additives also increased milk and solids yield of dairy cows during the hot season [13].

The similar result between treatments on the feed efficiency of concentrate supplement intake demonstrates that ADV was effective in promoting feed efficiency in beef cattle. As demonstrated by Duffield et al. [1], among the greatest benefits of using monensin is the ability to generate feed efficiency through reduced intake without affecting or increasing ADG. Although, in this study, a higher concentrate supplement consumption in the ADV treatment in relation to MON was observed, in the same way that ADV promoted higher ADG, a probable reason for the similarity on supplement G:F between treatments. Additionally, the ADV diet contained the trace mineral Cr which was absent in the MON treatment. It was previously demonstrated that the supplementary use of a zinc amino acid complex, in association with chromium methionine, improved feed efficiency of Nellore

bulls finished in confinement, compared with zinc amino acid supplementation alone or control treatment with all trace minerals from inorganic sources [12]. Budde et al. [42], observed with Cr propionate supplementation in diets containing 90 mg/kg of DM of supplemental Zn from Zn hydroxychloride, greater final BW, ADG and hot carcass weight in feedlot steers. Little is known about Cr absorption and metabolism [43]; however, Cr is known to be an important trace mineral that is associated with glucose metabolism [43] and potentiates the action of insulin in insulin-sensitive tissues [44].

Although a greater ADG in live weight was observed with the use of ADV, the ADG in carcass and the final HCW was not different between the treatments, despite the numerical difference of 8.3% in carcass ADG, which generated the numerical difference of 9.4 kg in the final HCW. However, the trend towards greater LM area (5.8%) observed with the use of ADV, may be indicative of greater muscularity and commercial yield of the carcasses as a result of possible higher cold carcass weight [45]. However, the fat thickness was similar between treatments. Recent studies evaluating the use of yeast products (Live yeast and yeast fermentation product) have not shown effects on HCW, LM area and fat thickness [3,5,36]. On the other hand, in agreement with the present results, supplementation strategies with organic sources of trace minerals have shown benefits in generating greater LM area, without affecting fat thickness, as with zinc proteinate supplementation to the detriment of zinc oxide [46], and with zinc amino acid complex and chromium methionine [12].

The similarity between the treatments on the liver concentrations of Cu, Mn and Se, indicate that the lower supplementary inclusion levels of trace minerals from organic sources present in the treatment with ADV was sufficient to maintain similar hepatic concentrations of these minerals compared to higher inorganic inclusion levels. It is worth mentioning that in the ADV treatment, the supplementary amount of these minerals was, respectively, 55.5, 29.2 and 33.3% lower for Cu, Mn and Se minerals in relation to the inorganic sources, whose supplementary amount of these minerals was established to meet or exceed NASEM recommendations [8]. This result demonstrates the possibility of reducing the level of supplementation with these trace minerals with the use of ADV, without a negative effect on the status of these minerals in cattle during a finishing period with high concentrate, since the liver is the organ that often represents the status of several trace elements in animals [47]. This result was probably due to the recognized higher bioavailability of organic source of trace minerals, with the possibility of generating greater absorption, retention and biological activity compared to inorganic minerals [9–11]. Additionally, the hepatic concentrations of the minerals evaluated in this study, in both treatments, are considered to be within adequate ranges [8,48,49].

The trend of lower zinc concentration observed In ADV was probably due to a lower supplementary amount of Zn (64.9% less than in the supplementation with inorganic sources) in relation to the inorganic sources of trace minerals. Nonetheless, the maintenance of liver Zn concentrations within the range established as adequate [8,48] and the greater ADG and LM area observed with the use of ADV demonstrates that the supplemental amount of Zn from zinc proteinate was sufficient to support the demand for tissue growth. Interestingly, Niedermayer et al. [50] observed that, at the end of a period of 125 days of finishing beef steers, a tendency of lower concentration of Zn in the liver of steers receiving 100 mg/kg of supplemental Zn (industry recommendation of trace minerals) in relation to steers receiving 30 mg/kg supplemental Zn [8], from zinc sulfate. However, supplementation of 100 mg/kg generated higher ADG and HCW [50]. Carmichael-Wyatt et al. [51], supplementing Zn at five times current NASEM recommendations [8], had no effect on liver Zn concentrations, supporting the assertation that liver Zn is insufficiently sensitive to distinguish among cattle with adequate or greater Zn status.

In the present study, there was no difference in chemical composition of the meat from bulls fed with ADV or MON. There was also no difference in most meat quality traits, except for the L* and b* values, which were greater for bulls fed ADV than for those fed MON. The greater lightness observed with the use of ADV suggests the benefit of

producing less dark meat, which is more attractive to the consumer, as this is an important attribute in the visual evaluation of meat [16]. Greater yellowness observed in meat has been associated with increased fat deposition [52]. Differences observed in meat lightness may be related to factors such as changes in the concentration of heme pigment, which can be generated due to a change in muscle fiber types and oxidative metabolism [53]. However, this hypothesis still needs to be investigated in relation to the treatments evaluated here. The backfat thickness, which is also an attribute that can influence meat L*, due to the effect of protecting the carcass during cooling during the postmortem period, was similar between MON and ADV and greater than 3 mm, which is a threshold value reported to effectively protect the carcass during cooling [54]. The trend towards higher carcass pH observed with the use of ADV may also be a factor influencing meat L*, however in the opposite direction, as higher pH has been correlated with lower L* [16]. Nonetheless, the pH observed in both treatments was below 5.8, which is considered for the meat industry as adequate to generate positive attributes in the meat [55]. In addition, Vestergaard et al. [53] reported differences in meat L* even at a similar meat pH, with darker meat color for bulls fed roughage compared to concentrate. According to Moholisa et al. [56], tenderness is one of the most important meat characteristics related to consumer satisfaction. In the present study, the Warner–Bratzler Shear Force was similar across treatments. The observed values, being greater than 42.8 kg, which was considered an upper limit to classify beef as tender [57], are probably due to the age of the animals used in this study [58].

## 5. Conclusions

Sodium monensin and inorganic sources of trace minerals were able to be removed from the diet of beef cattle finished on pasture with high concentrate supplementation, without impairing feed efficiency, and with benefits on animal growth and meat color, when a blend of live yeast and organic trace minerals were included in the diet. The greater ADG observed is likely be related to the higher intake of concentrate and higher digestibility of the diet with the use of live yeast and organic minerals. The use of trace minerals from organic sources, even with reduced supplementary levels, made it possible to maintain adequate mineral status during the finishing period. This study contributes to the current knowledge about the use of technologies in feed additives for beef cattle, demonstrating that the blend of live yeasts and organic minerals is an interesting nutritional alternative. However, further studies are required to evaluate the effects of this technology on ruminal metabolism and the mechanisms by which it was possible to improve meat color attributes.

**Author Contributions:** Conceptualization, M.S.S., L.F.C.e.S., A.K., V.H., G.R.S. and F.D.d.R.; methodology, M.S.S., L.H.C.B. and I.M.O.; validation, L.H.C.B. and M.S.S.; formal analysis, M.S.S., L.H.C.B. and I.M.O.; investigation, M.S.S., H.A.S.I., I.A.C. and I.M.F.; resources, A.K., V.H., G.R.S. and F.D.d.R.; data curation, M.S.S., G.R.S., F.D.d.R. and L.F.C.e.S.; writing—original draft preparation, L.H.C.B.; writing—review and editing, L.H.C.B., L.F.C.e.S., A.K., V.H., G.R.S., I.M.F. and I.A.C.; project administration, F.D.d.R., G.R.S. and I.M.O.; funding acquisition, F.D.d.R. and G.R.S. All authors have read and agreed to the published version of the manuscript.

**Funding:** This research was funded by Alltech Brazil (Maringá, PR, Brazil). The scholarship of the second author was supported by the National Council of Technological and Scientific Development (CNPq, Brazil #141577/2020-7).

**Institutional Review Board Statement:** This study was carried out in accordance with the ethical principle established by the Brazilian Council for the Control of Animal Experimentation and approved by the Ethics Committee for the Use of Animals of the Sao Paulo State University, College of Agricultural and Veterinary Sciences, Sao Paulo, Brazil (process no. 014782/19).

**Data Availability Statement:** Not applicable.

**Acknowledgments:** The authors acknowledge the support from the zootechnician Hugo Aparecido Silveira Issa, for his contribution in the execution of this study, in the stages of data collection and laboratory analysis.

**Conflicts of Interest:** Some of the co-authors are associated with the manufacturer of the feed additive tested.

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
