# Peer review of "Effects of a Blend of Live Yeast and Organic Minerals as an Alternative to Monensin on Intake, Digestibility, Performance and Beef Quality of Nellore Bulls Finished on Pasture with High Concentrate Supplementation"

_agriculture, doi:10.3390/agriculture13030522_

Round 1

Reviewer 1 Report

Review comments on “Effects of a blend of live yeast and organic minerals as an alternative to monensin on intake, digestibility, performance and beef quality of Nellore bulls finished on pasture with high concentrate supplementation” by Luis Henrique Curcino Batista etl.

This work presents and discusses the study Effects of a blend of live yeast and organic minerals as an alternative to monensin and inorganic trace minerals for beef cattle finished on pasture with high concentrate supplementation, on growth performance, intake, digestibility, liver trace mineral and carcass characteristics.

My main general comments are as below:

- An important shortcoming is that the author does not highlight the contribution of their manuscript in comparison to the work that has been performed by previous researchers. this can be added in the introduction and/or conclusion section.

-The methods do not describe the operation of the system for monitoring the nutrition of animals. What is the system for monitoring individual feeding behavior based on? what is the benefit of using this system in the described experiments.

-Of course, the topic of using live yeast in the nutrition of agricultural animals is currently very relevant among researchers, at the same time, the use of monensin in animal nutrition has been comprehensively studied by a large number of authors. It is not clear why the authors of the article did not include in the experiments the control group of animals kept on a diet without any additives, but only compared yeast (and trace elements) and monensin. The presence of a control group would show not only relative changes in productive indicators between the experimental groups, but also absolute ones, which is no less important.

- Conclusions need more elaboration about: outcomes, limitations, and possible/future scenarios.

- There is no symbol definition × in formulas.

- The authors should pay special attention to the design of formulas, numeration and notation.

- This manuscript cited too many not highly relevant references to the reported study, but on the other side not enough highly relevant references on the particular study the author(s) presented. Please correct this issue by removing less relevant references but adding a few (more) highly agricultural applications relevant references, to keep the total number of references around 50.

Reviewer 2 Report

The document entitled "Effects of a blend of live yeast and organic minerals as an alternative to monensin on intake, digestibility, performance and beef quality of Nellore bulls finished on pasture with high concentrate supplementation" has some novelty. The authors' should be commended on the depth of research conducted and the numerous analytical tools used in this project. The only major concern of this document is present in the methodology where there were no comparisons presented on the chemical and physical characteristics in which the animals grazed during the experiment. What was presented doesn't show this. If this is fixed and it can be shown that there is no statistical significance in the parameters the manuscript can be published.

Future work can involve applying dietary treatment to cattle earlier in life (maybe at weaning). As was stated in the manuscript there was a tendency (p 0.06) to have an increase in ADG. (Suggested work). We might even notice changes in the performance and carcass parameters.

Specific comments:

In the Methodology: If sampling of the two paddock were performed for comparison it would have been of more value to the paper. If this can be added to the document when any difference in the nutritive value of the forage at both locations can be identified to ensure that there is no variability in the forages which is assumed to be a constant. Similar to what was done for the the concentrate with supplements for the two treatment groups. What is also of some concern is the sharp rise in CP in the last part (112 days) of the forage in the experiment. 

Line 24: Scientific name of the yeast should be italicized.

The abbreviation 'LM' should be provided (in the first instance) before it is used throughout the manuscript. This was present in the Abstract and Introduction.

In the method no mentioned was made with statistical analysis being applied to the liver, carcass and meat parameters of the various groups. This should be inserted.

Reviewer 3 Report

I revised the manuscript entitled "Effects of a blend of live yeast and organic minerals as an alternative to monensin on intake, digestibility, performance and beef quality of Nellore bulls finished on pasture with high concentrate supplementation". The manuscript addressed an original and relevant topic, aiming to replace monensin in supplements for finishing bulls. The experimental design is well described; however, details about the methods and different parameters calculation should be provided to make the manuscript more understandable.

Major comments

Authors should provide details on the characteristics of the pasture. Lines 85-86. How was this pasture established? How many years before your experiment? In which season was the experiment conducted? At which growth step were the grasses?

The methods used for feed intakes lack justification; authors could provide some references to similar studies that use the same approach. Moreover, in lines 156-157: Please provide details, mainly references. Lines 180-186: It is important to justify the calculation for DMI estimation, providing a reference.

Minor comments

The title is too long

The summary is missing.

Please avoid using abbreviations in the abstract.

Lines 19-20. “assigned to 1 of 2 treatments” assigned to one of the two experimental diets.

Line 61. What does LM represent?

Line 106. This table should include the mineral composition of the diets. So the authors will avoid adding it to the footnote of so many tables.

Lines 281-294. You use (p = 0.73) and sometimes (p < 0.01). Please use the same presentation (p < 0.01) in the document. NDF (p ≥ 0.53): this is not a good way to write significance levels. Please correct it in the document.

I suggest the authors shorten the discussion.

Round 2

Reviewer 2 Report

The authors have addressed all the issues outline. It can be pushed to further processing. 

Reviewer 3 Report

The manuscript was significantly improved.